# Estimation of Photosynthetic Induction Is Significantly Affected by Light Environments of Local Leaves and Whole Plants in *Oryza* Genus

**DOI:** 10.3390/plants13121646

**Published:** 2024-06-14

**Authors:** Zhuang Xiong, Jian Xiao, Jinfang Zhao, Sicheng Liu, Desheng Yang, Dongliang Xiong, Kehui Cui, Shaobing Peng, Jianliang Huang

**Affiliations:** National Key Laboratory of Crop Genetic Improvement, Ministry of Agriculture Key Laboratory of Crop Ecophysiology and Farming System in the Middle Reaches of the Yangtze River, College of Plant Science and Technology, Huazhong Agricultural University, Wuhan 430070, China; xiongz@mail.hzau.edu.cn (Z.X.); cindo@webmail.hzau.edu.cn (S.L.); dlxiong@mail.hzau.edu.cn (D.X.); cuikehui@mail.hzau.edu.cn (K.C.); speng@mail.hzau.edu.cn (S.P.)

**Keywords:** photosynthetic induction, stomatal kinetics, light intensity, stomatal morphology, fluctuating light

## Abstract

Photosynthetic induction and stomatal kinetics are acknowledged as pivotal factors in regulating both plant growth and water use efficiency under fluctuating light conditions. However, the considerable variability in methodologies and light regimes used to assess the dynamics of photosynthesis (*A*) and stomatal conductance (*g*_s_) during light induction across studies poses challenges for comparison across species. Moreover, the influence of stomatal morphology on both steady-state and non-steady-state *g*_s_ remains poorly understood. In this study, we show the strong impact of IRGA Chamber Illumination and Whole Plant Illumination on the photosynthetic induction of two rice species. Our findings reveal that these illuminations significantly enhance photosynthetic induction by modulating both stomatal and biochemical processes. Moreover, we observed that a higher density of smaller stomata plays a critical role in enhancing the stomatal opening and photosynthetic induction to fluctuating light conditions, although it exerts minimal influence on steady-state *g*_s_ and *A* under constant light conditions. Therefore, future studies aiming to estimate photosynthetic induction and stomatal kinetics should consider the light environments at both the leaf and whole plant levels.

## 1. Introduction

The world population is rapidly increasing and is projected to surpass nine billion by 2050 [1,2]. Alongside this growth, the frequency and intensity of extreme weather events such as high temperatures, droughts, and waterlogging are on the rise due to global climate change [3,4,5]. Ensuring food security worldwide in the face of these challenges requires enhancing the output of staple crops like rice and other cereals. Photosynthesis has long been a target for improving yield potential, as it forms the basis of crop biomass [6]. While many studies have investigated the mechanisms of leaf photosynthetic efficiency under stable light conditions [7,8,9], plants are often confronted with fluctuating light levels due to factors like sun angle variations, passing clouds, and the movement of overlapping leaves under natural environments [10,11]. The ability of a plant to adapt to such fluctuations importantly impacts its daily carbon uptake and water use efficiency [12,13]. Recent research has increasingly focused on photosynthetic efficiency and stomatal kinetics under fluctuating light conditions [14,15,16]. However, differences in the light regimes employed across studies, in which photosynthetic photon flux density (PPFD) inside the leaf chamber were set at 0, 10, 50, or 100 μmol m^−2^ s^−1^ during initial light induction and measurements were conducted at darkness, low light, or natural condition, poses challenges for comparing results [17,18,19,20,21]. Here, we investigated the impact of light intensity within the leaf chamber of the infrared gas analyzer (IRGA Chamber Shading and Illumination) and the external environment (Whole Plant Shading and Illumination) on dynamic photosynthesis during light fluctuations.

When a shaded leaf is exposed to light, the photosynthetic rate gradually increases until it reaches a steady state, which is a process known as “photosynthetic induction” [14]. Previous studies have shown significant variations in photosynthetic induction among rice genotypes [21,22]. In general, the plants with faster photosynthetic induction can achieve a higher biomass accumulation [23,24]. During light induction, the photosynthetic rate is mainly limited by three phases: the activation of the electron transport rate, Rubisco carboxylation, and stomatal opening [25,26,27,28]. There is a continued debate regarding the key limitation of photosynthetic induction. For instance, Acevedo-Siaca et al. (2020) showed that Rubisco carboxylation is the primary limitation of photosynthetic induction [18]. However, Yamori et al. (2020) suggested that higher stomatal conductance (*g*_s_) contributes to the rapid adjustment of photosynthetic rate under fluctuating light conditions [29]. Actually, these two factors interact with each other; higher *g*_s_ can enhance the faster recovery of intercellular CO_2_ concentration (*C*_i_) during induction, subsequently contributing to the activation of Rubisco carboxylation [17,23]. Moreover, it is still unknown whether stomatal and biochemical limitations are influenced by IRGA Chamber Illumination and Whole Plant Illumination during photosynthetic induction.

Stomata are small pores on leaf surfaces, bordered by two pairs of guard cells, controlling CO_2_ uptake and H_2_O loss. Therefore, stomatal aperture plays a crucial role in determining plant photosynthesis and transpiration, responding to both environmental and internal signals [30,31,32]. Stomatal characteristics, such as density and size, are considered to correlate with photosynthetic efficiency under fluctuating and constant light conditions, determining maximum potential stomatal conductance (*g*_s,max_) [33,34,35]. Most studies suggest that a higher density of smaller stomata can support faster *g*_s_ kinetics in response to environmental fluctuations and higher *g*_s_ under constant conditions, thereby enhancing photosynthetic efficiency [36,37,38]. However, Zhang et al. (2019) observed an antagonistic relationship between stomatal size and *g*_s_ kinetics [17]. Moreover, Xiong et al. (2017) and Eyland et al. (2021) suggested that stomatal density and size may not be the key factors affecting *g*_s_ kinetics [39,40]. Therefore, further clarification is still necessary.

In this study, we selected two rice genotypes with contrasting stomatal morphology to test the following hypothesis: (1) the extent to which IRGA Chamber Illumination and Whole Plant Illumination influence rice photosynthetic induction; (2) the impact of IRGA Chamber Illumination and Whole Plant Illumination on stomatal and biochemical limitations; and (3) the potential relationship between stomatal morphology and steady- versus non-steady-state *g*_s_ under constant and fluctuating light conditions.

## 2. Materials and Methods

### 2.1. Plant Growth

Two wild rice genotypes, *Oryza australiensis* Domin. and *Oryza officinalis* Wall. ex Watt, which showed a significant difference in stomatal morphology, were selected for this study. *O. australiensis* possesses a significantly larger stomatal size and lower stomatal density than *O. officinalis*. On 11 April 2022, both the wild rice genotypes were planted outdoors on the campus of Huazhong Agricultural University. Two months later, the small tillers of *O. australiensis* and *O. officinalis* were cut off and transplanted into 3 L pots at a density of two tillers per pot in June 2022. Each pot was filled with 3 kg field paddy soil, and there were a total of four replications for each genotype in this experiment. Prior to transplanting, 2 g per pot of compound fertilizer containing 17% of N, P, and K, respectively, was applied as basal fertilizer. Additionally, 2 g of compound fertilizer was applied as a top-dressing fertilizer after 35 days of transplanting at the tillering stage. To prevent drought stress, the water level was maintained at a minimum of 2 cm above the soil surface in each pot. All the pots were randomly positioned outdoors.

### 2.2. Gas Exchange Measurements

Gas exchange measurements were performed using a portable gas exchange system (Li-6800; LI-COR, Lincoln, NE, USA) equipped with a 6 cm^2^ chamber on the youngest fully expanded leaves. All the measurements were conducted in a lab room with regulated air conditioning, humidification, and artificial lighting. Two different light intensities were applied at the whole plant level during the measurements. The uppermost leaves were exposed to artificial light sources maintaining a photosynthetic photon flux density (PPFD) of approximately 10 µmol m^−2^ s^−1^ under Whole Plant Shading (WPS) conditions, or 1000 µmol m^−2^ s^−1^ on Whole Plant Illumination (WPI) conditions. Inside the cuvette of the gas exchange system, the light intensity was controlled by Li-6800, with the PPFD set at 10 and 100 µmol m^−2^ s^−1^ for IRGA Chamber Shading (ICS) and IRGA Chamber Illumination (ICI), respectively, during the initial phase of light induction, respectively. The flow rate, CO_2_ concentration, relative humidity, and ambient temperature inside the leaf chamber were set to 500 µmol s^−1^, 400 µmol mol^−1^, 60–65%, and 28 °C, respectively.

During the measurements, the leaves were first acclimated to the low light levels inside the chamber, with the PPFD set at 10 μmol m^−2^ s^−1^ for IRGA Chamber Shading and 1000 µmol m^−2^ s^−1^ for Whole Plant Illumination. Then, the PPFD inside the leaf chamber was increased to 1500 µmol m^−2^ s^−1^ for 750 s of light induction. Once induction was complete, the PPFD was decreased to 100 µmol m^−2^ s^−1^ until the net photosynthetic rate (*A*) g_s_ reached a steady state. Following this, the PPFD inside the leaf chamber was increased again to 1500 µmol m^−2^ s^−1^ for another 750 s of light induction. This process was repeated under Whole Plant Shading conditions, with the PPFD in the room set at 10 µmol m^−2^ s^−1^. *A* and *g*_s_ were recorded every 6 s. *AC*_i_ response curves were conducted to calculate the maximum carboxylation rate (*V*_cmax_), maximum electron transport rate (*J*_max_), and triose phosphate use (TPU). During these measurements, the PPFD was initially set to 1500 umol m^−2^ s^−1^ and the concentration of CO_2_ was set to 400 µmol mol^−1^ in the leaf cuvette. Once *A* and *g*_s_ stabilized, an automated program was initiated, sequentially changing the CO_2_ concentration in the leaf cuvette to 400, 300, 200, 100, 50, 400, 400, 600, 800, 1000, 1200, 1500, and 2000 µmol mol^−1^. For each record, a minimum of 2 min and maximum 3 min wait were carried out.

### 2.3. Analysis of Leaf Photosynthetic Induction

To evaluate the response of *A* to a stepwise increase in light intensity, P_50*A*_ and P_90*A*_ were calculated, representing the time required for *A* to increase by 50% and 90% during the light induction, respectively. *g*_s_ response was simulated using an analytic model proposed by Vialet-Chabrand et al. (2017) [41]:$$g_{s}=\left(g_{sf}-g_{si}\right)e^{{-e}^{\left(\frac{\lambda -t}{k_{i}}+1\right)}}+g_{si}$$
where *g*_si_ and *g*_sf_ represent the initial and final *g*_s_ during light induction, *k_i_* is the time constant for the increase in *g_s_*, and *λ* is the initial lag time. The relative rate of *g*_s_ response (P_50*g*_ and P_90*g*_) was calculated similarly to P_50*A*_ and P_90*A*_.

Stomatal and biochemical limitations were calculated after Kaiser et al., (2017) and Acevedo-Siaca et al., (2020) [18,42]. To eliminate the diffusional limitations, the photosynthetic rate was normalized as *A*_Ca_* and *A*_Ci_*:$$A_{C_{a}}^{*}=A\cdot \frac{min\left\{A_{c}\left(C_{a}\right),\;{\;A}_{j}\left(C_{a}\right),\;A_{t}(\left.C_{a})\right\}\;\right.}{min\left\{A_{c}\left(C_{cif}\right),\;{\;A}_{j}\left(C_{cif}\right),\;A_{t}(\left.C_{cif})\right\}\;\right.}$$
$$A_{C_{i}}^{*}=A\cdot \frac{min\left\{A_{c}\left(C_{i}\right),\;{\;A}_{j}\left(C_{i}\right),\;A_{t}(\left.C_{i})\right\}\;\right.}{min\left\{A_{c}\left(C_{cif}\right),\;{\;A}_{j}\left(C_{cif}\right),\;A_{t}(\left.C_{cif})\right\}\;\right.}$$
where *A*_c_, *A*_j,_ and *A*_t_ were calculated using the FvCB model [43] modified to account for *TPU* limitation [44]. *C*_a_ and *C*_if_ represent the CO_2_ concentration in the chamber and the final intercellular CO_2_ concentration during the light induction, respectively.

The stomatal (LS) and biochemical limitation (LB) during the photosynthetic induction were calculated as follows:$$LS=\frac{A_{C_{a}}^{*}-A}{A_{f}-A_{i}}\;*\;100$$
$$LB=\frac{A_{f}-A_{C_{i}}^{*}}{A_{f}-A_{i}}\;*\;100$$
where *A*_i_ and *A*_f_ were the initial and final photosynthetic induction during the light induction, respectively.

### 2.4. Stomatal Morphology Traits

The stomatal morphology was measured using dental resin and nail varnish, following Weyers and Johansen (1985) [45]. The impressions of the abaxial and adaxial surfaces of the middle region of the leaves were taken after completing the gas exchange measurements. Four replications from different pots were measured. Four micrographs of the impressions were captured at ×200 and ×600 magnification using an optical microscope equipped with a digital camera (Nikon-Eclipse-Ti, Tokyo, Japan). The stomatal density and stomatal size were analyzed using the ImageJ software (Wayne Rasband/NIH, Bethesda, MD, USA). The stomatal size was calculated by multiplying the stomatal length and width.

The maximum leaf stomatal conductance to water vapor (*g*_s,max_) was calculated from the stomatal density and morphology according to Franks and Beerling (2009) [33]:$$g_{s,\;max\;}=\frac{d\times D\times a_{max}}{1.6v(l+\frac{\pi }{2}\sqrt{a_{max}/\pi \;})}$$
where *d* is the diffusive of water vapor in the air (m^2^ s^−1^); *v* is the molar volume of air (m^3^ mol^−1^); *D* is the stomatal density (no. m^−2^); *l* is the pore depth (μm); *a_max_* is the maximum pore area (μm^2^).

### 2.5. Statistical Analysis

The one-way and two-way analysis of variance (ANOVA) and the least-significant difference (LSD) test were performed to assess the impact of IRGA Chamber Illumination and Whole Plant Illumination on leaf photosynthetic induction using SPSS 21.0 (SPSS for Windows, Chicago, IL, USA). The graphs were generated using SigmaPlot 12.5 (Systat Software Inc., San Jose, CA, USA).

## 3. Results

### 3.1. Response of Gas Exchange Parameters to a Stepwise Increase in Light Intensity

After illumination, both *A* and *g*_s_ increased rapidly in all the treatments (Figure 1). During the light induction, *A* and *g*_s_ were remarkably higher for IRGA Chamber Illumination (WPI-ICI, WPS-ICI) compared to IRGA Chamber Shading (WPI-ICS, WPS-ICS), likely due to the lower initial values and slower induction rates when shaded (Table 1 and Table 2). Additionally, Whole Plant Illumination appeared to significantly influence the photosynthetic induction and stomatal kinetics during the measurements, as the photosynthetic induction and stomatal kinetics were faster for WPI-ICS compared to WPS-ICS (Figure 1 and Table 1). However, little difference in the photosynthetic induction and stomatal kinetics was observed between WPI-ICI and WPS-ICI. Our data showed that intercellular CO_2_ concentration (*C*_i_) was higher during the initial phases and then decreased rapidly after switching to high light conditions in all the treatments (Figure 2A,B). Conversely, intrinsic water use efficiency (*WUE*_i_) was lower during the initial phases and increased rapidly after switching to high light conditions in all the treatments (Figure 2C,D). Little effect of Whole Plant Illumination on *C*_i_ was observed. Conversely, *C*_i_ was lower during the initial phases and higher at the end of the induction for IRGA Chamber Illumination (ICI) compared to IRGA Chamber Shading (ICS) (Figure 2A,B).

To evaluate the effect of IRGA Chamber Illumination (ICI) and Whole Plant Illumination (WPI) on stomatal kinetics and photosynthetic induction, we calculated the response rate of stomatal opening and photosynthesis to light fluctuations (Table 1). We observed significantly lower P_50_ and P_90_ of *A* and *g*_s_ in IRGA Chamber Illumination (ICI) compared to IRGA Chamber Shading in both rice genotypes. Whole Plant Illumination decreased the P_50_ and P_90_ values of stomatal kinetics and photosynthetic induction under IRGA Chamber Shading (ICS) conditions, but increased them under IRGA Chamber Illumination (ICI) conditions. Additionally, WPS-ICS exhibited the slowest photosynthetic induction and stomatal opening (Table 1). These results suggest that IRGA Chamber Illumination (ICI) during the initial phase could enhance photosynthetic induction, while Whole Plant Illumination (WPI) could also enhance photosynthetic induction only when the light intensity in the leaf chamber is extremely low (ICS) (Table 1).

### 3.2. Stomatal and Biochemical Limitations to Photosynthetic Induction

IRGA Chamber Illumination (ICI) and Whole Plant Illumination (WPI) exhibited significant effects on stomatal and biochemical processes during light induction (Figure 3). IRGA Chamber Illumination accelerated photosynthetic induction by reducing biochemical limitations in both rice genotypes (Figure 3C,D). However, the impact of Whole Plant Illumination on photosynthetic induction may also be influenced by the light intensity inside the leaf chamber. Whole Plant Illumination had minimal effect on biochemical processes under IRGA Chamber Illumination conditions, but decreased biochemical limitations under IRGA Chamber Shading conditions (Figure 3C,D).

### 3.3. Differences of Photosynthetic Efficiency across Two Rice Genotypes

According to the light induction measurements, we observed a higher *A* for *O. australiensis* compared to *O. officinalis* (Figure 1A,B). Additionally, we evaluated the steady-state gas exchange parameters of both rice genotypes, which aligned with dynamic processes (Appendix A). Further analysis indicated that *O. australiensis* exhibited greater efficiency in CO_2_ diffusion and biochemistry (Appendix A). To investigate the impact of stomatal morphology on CO_2_ diffusion, we measured the stomatal size and density of both rice genotypes (Figure 4). The results showed that *O. australiensis* had larger stomatal size and lower stomatal density, as well as slower stomatal opening and photosynthetic induction under fluctuating light conditions (Figure 4 and Table 1). The maximum leaf stomatal conductance (*g*_s,max_) calculated based on stomatal morphology showed no significant difference between *O. australiensis* and *O. officinalis* (Figure 4D). These results suggest that the difference in theoretical *g*_s,max_ is not consistent with *g*_s,steady_ under constant light conditions.

## 4. Discussion

### 4.1. IRGA Chamber Illumination and Whole Plant Illumination Significantly Enhanced Rice Photosynthetic Induction

In natural environments, light levels on leaf surfaces exhibit high spatial and temporal variability due to factors such as sun angles, cloud over, and shading from overlapping leaves [10]. As crop canopy becomes more complex and heterogeneous, there is a vertical gradient of light distribution within the canopy, with light fluctuations occurring more frequently and less predictably with increasing canopy depth [46,47]. Understanding how plants adapt to such dynamic light conditions is crucial for improving photosynthetic performance in field conditions [24,32]. However, there is considerable variation in the measurements of photosynthetic induction and stomatal kinetics under fluctuating light conditions, as the light intensity inside and outside the leaf chamber varies among studies [17,19,22]. Here, we investigated the effect of different light regimes on leaf photosynthetic induction and stomatal kinetics (Figure 1). The results indicated that IRGA Chamber Illumination (ICI) significantly increased photosynthetic induction and accelerated stomatal opening, possibly due to the higher initial *g*_s_ (Table 1 and Table 2). Similar results have already been reported by Guo et al., (2016) and Hou et al., (2015) [48,49].

In the natural condition, plants are often exposed to sunlight, though the light intensity fluctuates from seconds to minutes in the field. However, in several studies, whole plants have been placed in darkness during the measurements of photosynthetic induction and stomatal opening [18,21]. In fact, shaded environments typically occur within canopies under sunfleck conditions. Shimadzu et al. (2019) showed that a whole irradiated plant can exhibit faster photosynthetic induction than an individually irradiated leaf [50]. Similarly, photosynthetic induction and stomatal opening were faster under the Whole Plant Illumination (WPI) than the Whole Plant Shading (WPS) conditions in the IRGA Chamber Shading condition for both rice genotypes (Figure 1 and Table 1). This consistency may be attributed to the faster stomatal opening under whole irradiated conditions. Conversely, the leaves under Whole Plant Illumination (WPI) displayed slower stomatal opening and photosynthetic induction than those under Whole Plant Shading (WPS) under the IRGA Chamber Illumination conditions across both rice genotypes (Table 1), possibly due to the higher final stomatal conductance and photosynthetic rate of WPI-ICI compared to WPS-ICI. Overall, leaf photosynthetic induction and stomatal opening are significantly influenced by Whole Plant Illumination and IRGA Chamber Illumination, emphasizing the importance of considering the light environments of both local leaves and the whole plant in future studies.

### 4.2. Stomatal and Biochemical Processes Were Influenced by Variations in Both Inside and Outside Light Intensity

*A* is determined by both the diffusion of CO_2_ from the atmosphere to carboxylation sites and CO_2_ carboxylation in the chloroplast. In field environments, light represents one of the most dynamic environmental factors. A sudden change in light intensity can lead to a decoupling of *A* and *g*_s_, potentially causing a *g*_s_ limitation to *A,* as stomatal opening often lags behind the induction of photosynthetic processes [15]. Moreover, biochemical processes, including the activation of electron transport and Rubisco, can also pose a major limitation to *A* during the initial period of photosynthetic induction [51]. Consistent with these findings, we observed a pronounced biochemical limitation at the onset of high light exposure, resulting in higher intercellular CO_2_ concentration (*C*_i_) and lower intrinsic water use efficiency (*WUE*_i_) (Figure 2 and Figure 3). Previous studies have suggested that stomata can limit *A* by up to 6.5–24.3% during light induction [15]. Similar results were evident in our study, where stomatal opening accounted for 10% to 20% of the limitation to photosynthetic induction in *O. sativa* (Figure 3).

Further, we estimated the impact of Whole Plant Illumination (WPI) and IRGA Chamber Illumination (ICI) on stomatal and biochemical processes (Figure 3). IRGA Chamber Illumination reduced biochemical limitation but increased stomatal limitation to photosynthetic induction. These results may be attributed to a higher CO_2_ supply through stomata compared to the CO_2_ demand inside leaves (Figure 2). Whole Plant Illumination clearly decreased both stomatal and biochemical limitations to photosynthesis, especially in *O. officinalis*. However, we observed little difference in intercellular CO_2_ (*C*_i_) concentration between Whole Plant Illumination and Whole Plant Shading. It appears that the effects of Whole Plant Illumination and IRGA Chamber Illumination on leaf photosynthetic induction depended on different mechanisms. Overall, Whole Plant Illumination (WPI) and IRGA Chamber Illumination (ICI) can accelerate leaf photosynthetic induction by modulating stomatal and biochemical processes, but the specific reasons behind these effects deserve further investigation.

### 4.3. Effect of Stomatal Morphology on Steady- and Non-Steady-State g_s_

Previous studies have predominantly focused on elucidating the mechanisms of photosynthetic processes under constant light conditions. However, in natural settings, plant carbon assimilation and growth rely heavily on dynamic responses to fluctuations in CO_2_ uptake. *A* estimated from the light–response curves may not reflect the actual carbon assimilation status outdoors during the growing season. Despite efforts to establish links between steady-state and non-steady-state photosynthesis, little correlation between these two processes has been observed [20,21]. However, it has been suggested that a higher initial *g*_s_ may contribute to faster stomatal kinetics [17]. In the present study, we observed minimal differences in the initial and final *g*_s_ between the Whole Plant Illumination (WPI) and Whole Plant Shading (WPS) conditions (Table 2). Notably, the initial *g*_s_ under IRGA Chamber Illumination (ICI) was significantly higher than under IRGA Chamber Shading (ICS) (Table 2). These results further underscore the differential effects of Whole Plant Illumination and IRGA Chamber Illumination on leaf photosynthetic induction, suggesting distinct underlying mechanisms. Moreover, it is worth noting that plants are often exposed to high light conditions in the field, and shaded environments on the whole plant level are rare.

Leaf gas exchange parameters are considered to be associated with stomatal characteristics under fluctuating light conditions, as they determine leaf CO_2_ uptake and H_2_O evaporation [52]. Plants with a higher density of smaller stomata are capable of achieving a higher maximum potential stomatal conductance (*g*_s,max_) and, subsequently, higher photosynthetic capacity [33]. This aligns with our findings, as *O. officinalis* exhibited a higher density of smaller stomata and a higher *g*_s,max_ (Figure 4). However, *g*_s,steady_ was lower in *O. officinalis* than in *O. australiensis*. In fact, *g*_s,max_ cannot be reached under normal conditions, as stomata are not consistently fully open. These results suggest that *g*_s_ and *A* are unlikely to be significantly affected by stomatal morphology under constant light conditions. Moreover, it remains unclear whether stomatal kinetics and photosynthetic induction in response to light fluctuations are related to stomatal morphology. Generally, a higher density of smaller stomata exhibits a rapid response to environmental stimuli [53]. Consistently, we observed faster stomatal kinetics and photosynthetic induction in *O. officinalis* compared to *O. australiensis* (Table 1). Therefore, stomatal morphology, which determines *g*_s,max_, has little effect on steady-state *g*_s_, but strongly influences stomatal kinetics and photosynthetic induction under fluctuating light conditions.

## 5. Conclusions

Our findings demonstrate that Whole Plant Illumination (WPI) and IRGA Chamber Illumination (ICI) significantly impact photosynthesis by modulating stomatal and biochemical processes during light induction. IRGA Chamber Illumination accelerates photosynthetic induction by reducing biochemical limitations. Conversely, Whole Plant Illumination accelerates photosynthetic induction by decreasing both stomatal and biochemical limitations. These two light regimes may exert contrasting effects on photosynthetic induction. Therefore, future studies estimating photosynthetic induction should consider the light conditions at both the local leaf and whole plant levels. Moreover, we observed that stomatal morphology strongly influences stomatal kinetics and photosynthetic induction but has minimal impact on steady-state stomatal conductance (*g*_s,steady_) and photosynthetic rate (*A*). The current study demonstrates a significant influence of the light environments of local leaves and whole plants on photosynthetic induction, which will contribute to the improvement of the photosynthetic efficiency and yield in rice.

## Figures and Tables

**Figure 1 plants-13-01646-f001:**
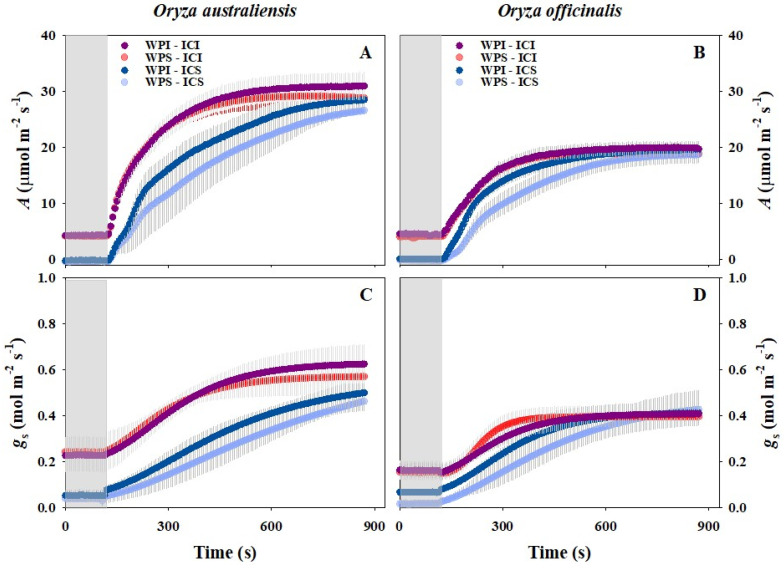
Response of photosynthetic rate (*A*) and stomatal conductance (*g*_s_) to a stepwise increase in irradiance (shade to white) across the two rice genotypes. WPI-ICI, WPS-ICI, WPI-ICS, and WPS-ICS represent measurements conducted under the Whole Plant Illumination and IRGA Chamber Illumination, Whole Plant Shading and IRGA Chamber Illumination, Whole Plant Illumination and IRGA Chamber Shading, and Whole Plant Shading and IRGA Chamber Shading conditions, respectively. The gray area (0–120 s) represents the initial phases at low light conditions for all measurements. The data presented are means (± SD) of all the treatments. The Standard Deviations (SDs) are shown in gray above and below the means for all the treatments.

**Figure 2 plants-13-01646-f002:**
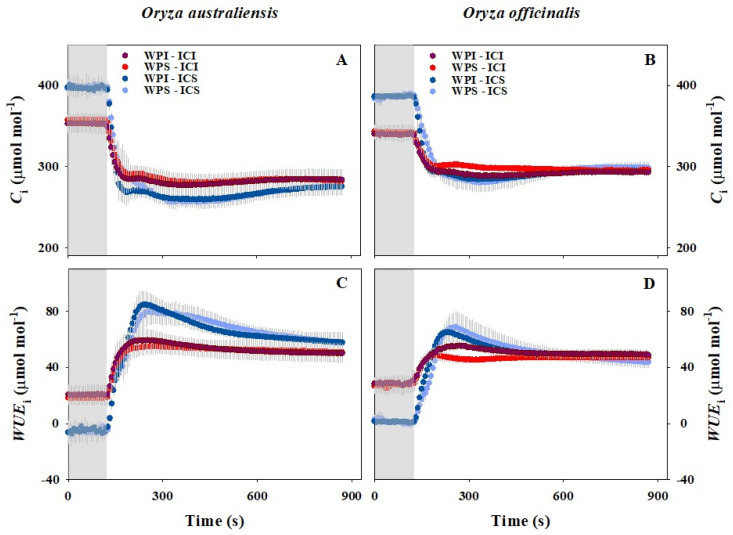
Response of intercellular CO_2_ concentration (*C*_i_) and intrinsic water use efficiency (*WUE*_i_) to a stepwise increase in irradiance (shade to white) across the two rice genotypes. WPI-ICI, WPS-ICI, WPI-ICS, and WPS-ICS represent the measurements conducted under the Whole Plant Illumination and IRGA Chamber Illumination, Whole Plant Shading and IRGA Chamber Illumination, Whole Plant Illumination and IRGA Chamber Shading, and Whole Plant Shading and IRGA Chamber Shading conditions, respectively. The gray area (0–120 s) represents the initial phases at low light conditions for all measurements. The data presented are means (± SD) of all the treatments. The Standard Deviations (SDs) are shown in gray above and below the means for all the treatments.

**Figure 3 plants-13-01646-f003:**
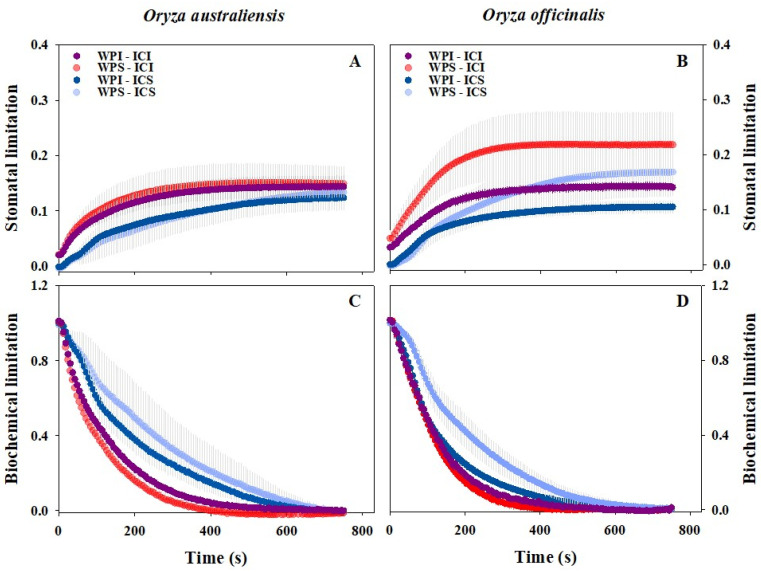
Transient stomatal and biochemical limitation to photosynthesis during light induction across the two rice genotypes. WPI-ICI, WPS-ICI, WPI-ICS, and WPS-ICS represent the measurements conducted under the Whole Plant Illumination and IRGA Chamber Illumination, Whole Plant Shading and IRGA Chamber Illumination, Whole Plant Illumination and IRGA Chamber Shading, and Whole Plant Shading and IRGA Chamber Shading conditions, respectively. The data presented are means (± SD) of the four measurements. The Standard Deviations (SDs) are shown in gray above and below the means for all the treatments.

**Figure 4 plants-13-01646-f004:**
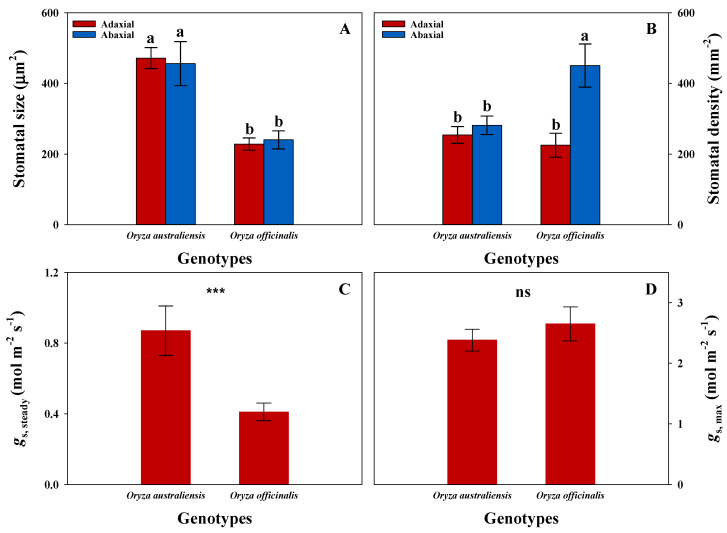
Comparison of stomatal morphology and stomatal conductance across the two rice genotypes. The adaxial and abaxial stomatal size and density are presented. *g*_s, steady_ is measured at a PPFD of 1500 µmol m^−2^ s^−1^ and CO_2_ concentration of 400 µmol mol^−1^. *g*_s,max_ is calculated based on stomatal morphology according to Franks and Beerling (2009) [33]. All the values are means (± SD) of the four measurements. Different letters indicate statistically significant differences (*p* < 0.05) across the two rice genotypes (**A**,**B**). Significance levels are indicated by *** and ns that indicate *p* < 0.001 and *p* > 0.05, respectively (**C**,**D**).

**Table 1 plants-13-01646-t001:** Calculated time required for *g*_s_ and *A* to increase by 50% (P_50*g*_, P_50*A*_) and 90% (P_90*g*_, P_90*A*_). WPI-ICI, WPS-ICI, WPI-ICS, and WPS-ICS represent measurements conducted under Whole Plant Illumination and IRGA Chamber Illumination, Whole Plant Shading and IRGA Chamber Illumination, Whole Plant Illumination and IRGA Chamber Shading, and Whole Plant Shading and IRGA Chamber Shading conditions, respectively. All values are means (± SD). Values marked with different letters are significantly different across treatments (*p* < 0.05).

Treatments	*Oryza australiensis*				*Oryza officinalis*			
	P_50*g*_ (s)	P_90*g*_ (s)	P_50*A*_ (s)	P_90*A*_ (s)	P_50*g*_ (s)	P_90*g*_ (s)	P_50*A*_ (s)	P_90*A*_ (s)
WPI-ICI	204 ± 39 bc	444 ± 74 bc	90 ± 5 b	312 ± 5 b	161 ± 34 bc	357 ± 71 b	98 ± 6 b	279 ± 20 ab
WPI-ICS	288 ± 68 ab	567 ± 88 ab	149 ± 35 ab	485 ± 61 a	198 ± 28 ab	432 ± 53 ab	99 ± 8 b	357 ± 18 a
WPS-ICI	168 ± 54 c	372 ± 117 c	74 ± 6 b	258 ± 34 b	116 ± 3 c	218 ± 20 c	93 ± 8 b	230 ± 10 b
WPS-ICS	343 ± 105 a	610 ± 84 a	215 ± 107 a	530 ± 84 a	272 ± 88 a	533 ± 136 a	173 ± 42 a	443 ± 65 a

**Table 2 plants-13-01646-t002:** Initial and final stomatal conductance and photosynthesis during light induction. WPI-ICI, WPS-ICI, WPI-ICS, and WPS-ICS represent measurements conducted under Whole Plant Illumination and IRGA Chamber Illumination, Whole Plant Shading and IRGA Chamber Illumination, Whole Plant Illumination and IRGA Chamber Shading, and Whole Plant Shading and IRGA Chamber Shading conditions, respectively. *g*_si_ and *A*_i_ are initial *g*_s_ and *A* during induction. *g*_sf_ and *A*_f_ are final *g*_s_ and *A* during induction. All values are means (± SD). Values marked with different letters are significantly different across treatments (*p* < 0.05).

Treatments	*Oryza australiensis*				*Oryza officinalis*			
	*g*_si_ (mol m^−2^ s^−1^)	*g*_sf_ (mol m^−2^ s^−1^)	*A*_i_ (µmol m^−2^ s^−1^)	*A*_f_ (µmol m^−2^ s^−1^)	*g*_si_ (mol m^−2^ s^−1^)	*g*_sf_ (mol m^−2^ s^−1^)	*A*_i_ (µmol m^−2^ s^−1^)	*A*_f_ (µmol m^−2^ s^−1^)
WPI-ICI	0.23 ± 0.07 a	0.63 ± 0.09 a	4.3 ± 0.2 a	30.9 ± 2.5 a	0.15 ± 0.04 a	0.41 ± 0.05 a	4.4 ± 0.2 a	19.7 ± 1.5 a
WPI-ICS	0.08 ± 0.02 b	0.50 ± 0.06 b	−0.2 ± 0.1 b	28.8 ± 1.4 a	0.08 ± 0.02 b	0.41 ± 0.03 a	−0.1 ± 0.1 b	19.6 ± 1.2 a
WPS-ICI	0.26 ± 0.08 a	0.58 ± 0.09 b	4.3 ± 0.1 a	28.8 ± 2.5 a	0.16 ± 0.04 a	0.40 ± 0.05 a	4.3 ± 0.3 a	18.9 ± 1.5 a
WPS-ICS	0.06 ± 0.04 b	0.50 ± 0.05 b	−0.2 ± 0.1 b	27.7 ± 2.1 a	0.03 ± 0.01 c	0.43 ± 0.08 a	0.0 ± 0.1 b	18.8 ± 1.7 a

## Data Availability

The datasets collected and/or analyzed in the present study are available from the corresponding author upon reasonable request.

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
