# Peer review of "Estimation of Photosynthetic Induction Is Significantly Affected by Light Environments of Local Leaves and Whole Plants in Oryza Genus"

_plants, 2024, doi:10.3390/plants13121646_

Round 1

Reviewer 1 Report (Previous Reviewer 1)

Comments and Suggestions for Authors

I have reviewed the authors' response and the revised manuscript. I am satisfied with the authors' answers to my comments and concerns. I believe that the information provided in their response should be included in the manuscript to enhance its clarity and comprehensiveness. 

Author Response

Dear reviewer,

We greatly appreciate the very constructive comments and suggestions from you , which would be very helpful in improving the quality of our manuscript. We have gone through all the comments carefully and made corrections.

Kind regards,

Zhuang Xiong

Reviewer 2 Report (New Reviewer)

Comments and Suggestions for Authors

Plants Manuscript #:  3034000

Authors: Z. Xiong et al., 2024

Title:  Estimation of photosynthetic induction is significantly affected by light environments of local leaf and whole plant in Oryza genus

The authors have presented key physiological data to assess photosynthetic induction and how different assessment methods and variations in light intensity used with these methods can significantly impact the resulting data and conclusions.  Further, they tested these methods on two different species/genotypes of rice (Oryza australiensis and Oryza officinalis) with differing stomatal size and density, for the impact on CO2/water conductance and photosynthetic induction.  The authors argue that better assessment and more consistent assessment methods that and factor in light intensity on the whole plant is needed in the industry to more accurately determine the important and very frequent effects of photosynthetic induction “in the field” that impacts total photosynthetic yield and biomass production.  Their methodology, statistical analysis and conclusions from data are sound.  The writing and clarity of information is also good, with a few minor edits/suggestions that I list below.

There are four general topics/sections/edits that I feel should be added to improve the manuscript, understanding by readers, and possible use to broader audience of plant scientist.  First, in the introduction (near Lines 50/51), add a sentence or two that would explain more generally the current methods used to assess photosynthetic induction so as to provide a clear distinction with what your (the authors’) methods are and that are recommended.  Second, near end of Introduction (Line 85) the authors say the two rice species/genotypes were chosen for the difference in stomatal morphology.  Since there was clear knowledge of differences at start of the study, the authors should state which species has more/larger stomata and which smaller/fewer (or at least general differences in stomatal conductance between the two).  Third, in the Conclusion (near Lines 377/378) add a sentence or two with specific and concrete examples for how the authors feel this information could be used to improve rice (and other plant) production and crop yields.  Forth, a formatting issue, use Numbered (#) references instead of the authors names/dates for the “in-text” citations.  Then, the reader can much more easily cross reference the sources in the Reference List at end of manuscript.  As is, it is very difficult, since Reference list has numbers and is in order of appearance, not alphabetical.

Summary, the general topic and methodological use of this information, as important as it is, is still fairly specialized.  But, in general, I support publishing it for it provides new information as well as recommendations for how to better assess environmental impacts on photosynthetic induction, once my recommendations above as well as some minor corrections/edits, which I list below, have been addressed.

Introduction:

Line 79:  In text citation is for “Woodward et al., 2003…” while in the Reference List it is “Hetherington AM, Woodward…”  Use the correct on in both locations.

Methods:

Line 126:  Please define “TPU”.

Results:

Line 181:  I would add “when shaded” at end of sentence, so that it would read, “…slower induction rates, when shaded.”

Figure 1:  Additional information about the figures is needed. This includes, in figure legend explain what the gray shaded area between 0 and ~100 sec is in all four panels.  Also, the +/- Standard Deviations (SD) are presumably shown with the “gray” bars above and below the colored lines (averages).  Please clarify that to the readers.

Figure 2:  Same questions/needs as mentioned above for Figure 1.

Lines 233/234:  The decrease in biochemical limit seems more pronounced around 200 seconds (? If significantly different) in Orzya officinallis for WPI-ICS vs WPS-ICS as compared to Orzya australiensis?  Is that the case? Is it significant? Any reasons why?

Figure S1 (Supplemental): In figure legend and on figure panels, please include p-value statistics to show if the observed differences between species/genotypes are significant or not.  It seems very likely some of the differences are statistically significant, as authors have done for other figures.

Line 246:  Regarding data in Figure 4D, the authors say there was a “slightly lower “gs, max” between species (4D), but from the figure and the authors statistics, these are not significantly different.  Thus, it seems incorrect to claim they are “slightly lower”, suggesting they are different.  I would remove that statement.  It does not undermine the next sentence, that there is a difference between “max” and “steady state”, under constant light.

Figure 4:  Legend needs an explanation of the “a” and “b” letters regarding statistical significant, as mentioned in Table 1, in panels A and B.

Table 2:  This “results” table was not mentioned, cited or discussed in the Results Section, as it should be.  Instead, it was first mentioned in the Discussion, Line 278.  Move Table 2 and it’s results interpretation to the Results Section.

Lines 278/279:  The authors mentioned that similar results have already been reported in Guo et al. (2016) and Hou et al. (2015).  So, what makes the authors study/results different? Worth publishing? 

Discussion:

Line 344:  Spelling typo, should be “minimal”

Reference List:

Lines 470/471: As mentioned above (Line 79), inconsistency between “in text” citation and Reference List for authors names for this reference.

Comments on the Quality of English Language

See my suggested edits above.

Author Response

The authors have presented key physiological data to assess photosynthetic induction and how different assessment methods and variations in light intensity used with these methods can significantly impact the resulting data and conclusions.  Further, they tested these methods on two different species/genotypes of rice (Oryza australiensis and Oryza officinalis) with differing stomatal size and density, for the impact on CO2/water conductance and photosynthetic induction.  The authors argue that better assessment and more consistent assessment methods that and factor in light intensity on the whole plant is needed in the industry to more accurately determine the important and very frequent effects of photosynthetic induction “in the field” that impacts total photosynthetic yield and biomass production.  Their methodology, statistical analysis and conclusions from data are sound.  The writing and clarity of information is also good, with a few minor edits/suggestions that I list below.

Response: We greatly appreciate the very constructive comments and suggestions from the reviewer, which would be very helpful in improving the quality of our manuscript. We have gone through all the comments carefully and made corrections. Please note that the modifications in the revised manuscript were highlighted in blue color.

There are four general topics/sections/edits that I feel should be added to improve the manuscript, understanding by readers, and possible use to broader audience of plant scientist.  First, in the introduction (near Lines 50/51), add a sentence or two that would explain more generally the current methods used to assess photosynthetic induction so as to provide a clear distinction with what your (the authors’) methods are and that are recommended.  

Response: Thank you for this helpful suggestion. We have rephased the sentence. Relevant descriptions can be found in line 41-45.

Second, near end of Introduction (Line 85) the authors say the two rice species/genotypes were chosen for the difference in stomatal morphology.  Since there was clear knowledge of differences at start of the study, the authors should state which species has more/larger stomata and which smaller/fewer (or at least general differences in stomatal conductance between the two). 

Response: Thanks for your comments. We have stated the stomatal characteristics of two rice genotypes in the M&M. Relevant descriptions can be found in line 85-88.

Third, in the Conclusion (near Lines 377/378) add a sentence or two with specific and concrete examples for how the authors feel this information could be used to improve rice (and other plant) production and crop yields.  

Response: Thanks for your suggestions. We have added the sentence in line 377-379.

Forth, a formatting issue, use Numbered (#) references instead of the authors names/dates for the “in-text” citations.  Then, the reader can much more easily cross reference the sources in the Reference List at end of manuscript.  As is, it is very difficult, since Reference list has numbers and is in order of appearance, not alphabetical.

Response: Thanks for your suggestions. We have revised all the format of literature citations.

Summary, the general topic and methodological use of this information, as important as it is, is still fairly specialized.  But, in general, I support publishing it for it provides new information as well as recommendations for how to better assess environmental impacts on photosynthetic induction, once my recommendations above as well as some minor corrections/edits, which I list below, have been addressed.

Response: We are very grateful to the reviewer for his or her appreciation of this work.

Introduction:

Line 79:  In text citation is for “Woodward et al., 2003…” while in the Reference List it is “Hetherington AM, Woodward…”  Use the correct on in both locations.

Response: Corrected.

Methods:

Line 126:  Please define “TPU”.

Response: Thanks for your suggestions. We have added the definition in line 120-121.

Results:

Line 181:  I would add “when shaded” at end of sentence, so that it would read, “…slower induction rates, when shaded.”

Response: Added.

Figure 1:  Additional information about the figures is needed. This includes, in figure legend explain what the gray shaded area between 0 and ~100 sec is in all four panels.  Also, the +/- Standard Deviations (SD) are presumably shown with the “gray” bars above and below the colored lines (averages).  Please clarify that to the readers.

Response: Thanks for your suggestions. We have added the description in line 193-196.

Figure 2:  Same questions/needs as mentioned above for Figure 1.

Response: Added.

Lines 233/234:  The decrease in biochemical limit seems more pronounced around 200 seconds (? If significantly different) in Orzya officinallis for WPI-ICS vs WPS-ICS as compared to Orzya australiensis?  Is that the case? Is it significant? Any reasons why?

Response: Thanks for your comments. It’s true that Orzya officinallis showed a larger decrease of biochemical limitation for WPI-ICS relative to WPS-ICS than Orzya australiensis, which might due to the genotype difference of biochemical processes in response to fluctuating light, and Orzya officinallis might have a faster activation of Vcmax and Jmax than Orzya australiensis after a stepwise increase in light intensity. I think this is also the reason why photosynthetic induction (P50A) between WPI-ICS and WPS-ICS showed a significant difference in Orzya officinallis, but not significant in Orzya australiensis (Table 1).

Figure S1 (Supplemental): In figure legend and on figure panels, please include p-value statistics to show if the observed differences between species/genotypes are significant or not.  It seems very likely some of the differences are statistically significant, as authors have done for other figures.

Response: Thanks for your suggestions. We have reanalyzed the data.

Line 246:  Regarding data in Figure 4D, the authors say there was a “slightly lower “gs, max” between species (4D), but from the figure and the authors statistics, these are not significantly different.  Thus, it seems incorrect to claim they are “slightly lower”, suggesting they are different.  I would remove that statement.  It does not undermine the next sentence, that there is a difference between “max” and “steady state”, under constant light.

Response: Thanks for your suggestions. We have made revision.

Figure 4:  Legend needs an explanation of the “a” and “b” letters regarding statistical significant, as mentioned in Table 1, in panels A and B.

Response: Added.

Table 2:  This “results” table was not mentioned, cited or discussed in the Results Section, as it should be.  Instead, it was first mentioned in the Discussion, Line 278.  Move Table 2 and it’s results interpretation to the Results Section.

Response: Thanks for your helpful suggestion. We have made revision.

Lines 278/279:  The authors mentioned that similar results have already been reported in Guo et al. (2016) and Hou et al. (2015).  So, what makes the authors study/results different? Worth publishing?

Response: Yes, involvement of systemic signal in photosynthetic induction have been reported by Guo et al. (2016) and Hou et al. (2015). However, this phenomenon doesn’t get attention yet, since an increasing numbers of scientists have been focused on photosynthetic induction after that, and the environments controlled during measurements varies widely among studies. Besides, the light intensity set in the leaf chamber posed also a significant effect on photosynthetic induction, which has not been reported up to now. Therefore, the current results may be beneficial for the study of photosynthesis under fluctuating light conditions, and improvement of photosynthetic efficiency and yield in rice.

Discussion:

Line 344:  Spelling typo, should be “minimal” b

Response: Corrected.

Reference List:

Lines 470/471: As mentioned above (Line 79), inconsistency between “in text” citation and Reference List for authors names for this reference.

Response: Corrected.

This manuscript is a resubmission of an earlier submission. The following is a list of the peer review reports and author responses from that submission.

Round 1

Reviewer 1 Report

Comments and Suggestions for Authors

Dear Authors,

I consider the manuscript to be written substantively and logically. I don't know if I missed this information, but could the authors of the paper explain why they chose for research these two wild rice genotypes ? Literature items recorded in the bibliography must be standardized in the entry.

Author Response

Reviewer: I consider the manuscript to be written substantively and logically. I don't know if I missed this information, but could the authors of the paper explain why they chose for research these two wild rice genotypes ? Literature items recorded in the bibliography must be standardized in the entry.

Response: Thanks for your comments. The reason we selected these two wild rice genotypes is that we observed a significant difference of stomatal morphology between these two wild rice genotypes in our preliminary experiments. Moreover, we have revised the format of the references in the literature.

Reviewer 2 Report

Comments and Suggestions for Authors

Overall I have two main issues with the paper as it stands, firstly it is not clear during the abstract or later on that these are wild relatives of Oryza species, not domesticated varieties. I think this is quite a large oversight and should have been a discussion point of the paper- with characterisation of these two species and potential interesting traits they could confer to domesticated varieties. A lot of the backing text and statement relate to agricultural crop canopies, and thus do not necessarily hold for wild relatives that aren’t cultivated in the same way. I also have a problem with the methods- why are tillers detached from plants for measurements and why not use whole intact plants. Particularly for biochemical and photosynthetic analysis.

Additional comments below. There is also a lot of grammatical errors not included in this list.

Abstract

Line 14- delete ‘are’, hand singular not plural

Line 16- it is not clear whether you are talking about IRGA versus whole plant illumination separately or combined at this point.

Introduction

Line 32- consider deleting ‘worldwide in the future’ from the end of this sentence. It reads better without

Line 36- consider changing ‘As a matter of fact(s)’ to ‘However…’

Line 38- the ability of a plant to adapt to….

Line 42- rephrase this sentence, it does not make sense

Line 51- shaded leaf, in a process called

Line 52- delete is

Line 63- interact with each other

Line 79- doesn’t this actually agree with the previous sentence…?

Methods

Line 92- why say the genus was chosen and not just name the two wild species. It should also be made clear in the abstract that this is wild Oryza species, not domesticated varieties.

Line 94- surely cutting off the tillers will alter the rice. Why don’t you just use whole plants? Or even whole plants with tillers removed so that they still have an intact root system.

Line 199-n is this paddy rice then?

Paragraph beginning line 117- repetition of WPS and WPI information

Line 125- how often did you match the IRGA chamber

Line 163- size assumed as a rectangle? Or as an ellipse?

Results

Figure 1- why are there only 2 lines when then are 4 different treatments?

You switch between the full treatment name and acronym throughout

Line 233- diffusion

Line 245- rewrite sentence

Comments on the Quality of English Language

There are a number of grammatical errors and typos throughout which requiring changing. 

Author Response

Reviewer:

Overall I have two main issues with the paper as it stands, firstly it is not clear during the abstract or later on that these are wild relatives of Oryza species, not domesticated varieties. I think this is quite a large oversight and should have been a discussion point of the paper- with characterisation of these two species and potential interesting traits they could confer to domesticated varieties. A lot of the backing text and statement relate to agricultural crop canopies, and thus do not necessarily hold for wild relatives that aren’t cultivated in the same way. I also have a problem with the methods- why are tillers detached from plants for measurements and why not use whole intact plants. Particularly for biochemical and photosynthetic analysis.

Response: Thanks for your comments. We have explained in the Materials and Methods part that the selected materials are two wild rice genotypes, which the changed part are highlighted in blue. I can’t more agree with the reviewer’s opinion that characteristics of these two wild genotypes should also be an important discussion point as their growth situation and cultivation pattern are completely different from domesticated genotypes. However, we focused mainly on the effect of two light regime on leaf photosynthetic induction and stomatal kinetics, and many agronomic traits are not measured in the present study. We will pay more attention on the difference of the growth and development across wild rice genotypes and domesticated genotypes in future research. What’s more, all the measurements are indeed conducted on intact leaves and  plants.

Additional comments below. There is also a lot of grammatical errors not included in this list.

Response: Thanks for your comments. We have carefully revised the grammar of the literature.

Abstract

Line 14- delete ‘are’, hand singular not plural

Response: Corrected.

Line 16- it is not clear whether you are talking about IRGA versus whole plant illumination separately or combined at this point.

Response: Corrected.

Introduction

Line 32- consider deleting ‘worldwide in the future’ from the end of this sentence. It reads better without

Response: Corrected.

Line 36- consider changing ‘As a matter of fact(s)’ to ‘However…’

Response: Corrected.

Line 38- the ability of a plant to adapt to….

Response: Corrected.

Line 42- rephrase this sentence, it does not make sense

Response: Corrected.

Line 51- shaded leaf, in a process called

Response: Corrected.

Line 52- delete is

Response: Corrected.

Line 63- interact with each other

Response: Corrected.

Line 79- doesn’t this actually agree with the previous sentence…?

Response: Yes, the results in Zhang et al. (2019) showed that smaller stomata have lower gsi  and slower initial response rate than larger ones in rice plants.

Methods

Line 92- why say the genus was chosen and not just name the two wild species. It should also be made clear in the abstract that this is wild Oryza species, not domesticated varieties.

Response: Corrected.

Line 94- surely cutting off the tillers will alter the rice. Why don’t you just use whole plants? Or even whole plants with tillers removed so that they still have an intact root system.

Response: Thanks for your comments. This is a method of transplanting wild rice, and the new root is retained on the small tillers.

Line 199-n is this paddy rice then?

Paragraph beginning line 117- repetition of WPS and WPI information

Response: Corrected.

Line 125- how often did you match the IRGA chamber

Response: Every 15-20 minutes.

Line 163- size assumed as a rectangle? Or as an ellipse?

Response: Rectangle.

Results

Figure 1- why are there only 2 lines when then are 4 different treatments?

You switch between the full treatment name and acronym throughout

Response: Thanks for your comments. There are four lines. We have revised the full treatment name and acronym throughout the literature.

Line 233- diffusion

Response: Corrected.

Line 245- rewrite sentence

Response: Corrected.

 Comments on the Quality of English Language

There are a number of grammatical errors and typos throughout which requiring changing.

Response: Thanks for your comments. We have carefully revised the grammar of the literature. 

Reviewer 3 Report

Comments and Suggestions for Authors

The manuscript entitled: Estimation of Photosynthetic Induction is Significantly Affected by Light Environments of Local Leaf and Whole Plant in Oryza genus showed the effect of low and high light to the two species of rice plants when the entire plant (WPI vs WPS) and/or the leaf section (ICI vs ICS) is acclimated prior to measurements of photosynthetic induction and CO2 response curve.

The study, having a neat experimental design, would provide useful observation on the effect of low vs high light exposure of either the entire plant or the leaf itself on the photosynthetic parameters that are usually investigated in plant physiological studies. This study provides promising results, however I believe the author has to do a better job in describing the results.

At the moment, what is described in the result section is either not visible in the figures, or sometimes even contrary to what the figure shows. For example, line 179 to 181 states that during light induction, A and gs were lower for WPI-ICI and WPS-ICI than WPI-ICS and WPS-ICS, but the contrary is what is shown in Figure 1. Also, in line 184-185, stomatal kinetics of WPI-ICS being significantly faster than that of WPS-ICS, is not visible in Figure 1. Both Figure 1 and 3 does not have figure legend for WPS-ICI and WPS-ICS. Furthermore, the error bars seem to be off, e.g. the max and min SD are not of the same length. Why is that? It is also not visible in some figures what stepwise increase in irradiance mean (shade to white) - please direct more the readers on this. 

I also suggest that the authors switch Figure 3 and Figure 4, as they first described Figure 4 in the Results section, this figure should be renamed as Figure 3 instead, and vice versa. This is common practice. 

During the discussion, I would suggest that the authors would first translate what the treatments mean in physiological terms (i.e. instead of Illumination and Shading, use low light and high light, respectively). This will create a more targeted discussion and would be more easy to understand by the readers. Moreover, once the authors have described the results more correctly, they can adjust their discussion later on. Currently, there are parts of the discussion that is not very visible in the results (e.g. line 346 to 347, line 351 to 352 and line 367 to 370). I suggest that the authors think more carefully of their results in terms of causality. Some of the observations does not necessarily mean that one treatment causes the plant to have less A or gs. If the authors would like to infer the results in causality, I suggest to cite more papers (stronger evidence) with mechanistic description of the observed relationships.

Other minor comments:

I suggest refining more the abstract, the hypothesis does not seem to be compatible with the method. Please refine the overall hypothesis by summarising the last 3 questions in the end of introduction, and bring that to the abstract.

Line 30, nice sentnce.

Line 36, use a better conjunction. Instead of "As a matter of fact", perhaps use, "However, in reality,"... This will connect sentence 34-35 to sentence 36-37

Line 39-40, needs reference

Line 46-47: IRGA chamber Shading is just low light condition in the method, I suggest to use low light instead of shading, as they would mean different things, especially with the spectral quality.

Line 59-60: Good content, but citation is confusing. Why cite Acevedo-Siaca et al 2021, if the RUBISCO carboxylation limitation is showed in their 2020 paper?

Line 66-68, I think there are already research whether stomatal and biochemical limitation affected by low and high light during photosynthetic induction, so we know a lot about it (in several crop species).

Line 94: put s in month

Line 117 to 131: I suggest to make a schematic diagram to illustrate the changes in light intensity for each treatment. This will give the reader a better grasp of the method.

Line 179-181, Figure 1 shows the contrary. Please be more careful with how you describe your results. This makes your discussion difficult to trust if your results are not correctly described.

Line 184 - 185, this is not visible in Figure 1, (same with line 186-187)

Line 191-192: It seems that lowe Ci at the ind of induction in ICI than ICS is not significant. Can you test?

Line 213-216, please point the reader to which Figure.

Line 216, I dont see this in Table 1. Can you please reassess?

Line 232-233: This was not indicated in the results or supplementary figures?

Comments on the Quality of English Language

I suggest editing the paper for english correction. Lots of parts of the manuscript had wrong grammar (for example, line 14, line 52, line 94, so on..)

Author Response

Reviewer:

The manuscript entitled: Estimation of Photosynthetic Induction is Significantly Affected by Light Environments of Local Leaf and Whole Plant in Oryza genus showed the effect of low and high light to the two species of rice plants when the entire plant (WPI vs WPS) and/or the leaf section (ICI vs ICS) is acclimated prior to measurements of photosynthetic induction and CO2 response curve.The study, having a neat experimental design, would provide useful observation on the effect of low vs high light exposure of either the entire plant or the leaf itself on the photosynthetic parameters that are usually investigated in plant physiological studies. This study provides promising results, however I believe the author has to do a better job in describing the results.

Response: We thank the reviewer for his or her positive comments, and appreciate the criticisms that were raised below, which were helpful for strengthening the current manuscript. The revised text has been indicated in blue color.

At the moment, what is described in the result section is either not visible in the figures, or sometimes even contrary to what the figure shows. For example, line 179 to 181 states that during light induction, A and gs were lower for WPI-ICI and WPS-ICI than WPI-ICS and WPS-ICS, but the contrary is what is shown in Figure 1.

Response: Thanks for your comments. We have revised the sentence.

Also, in line 184-185, stomatal kinetics of WPI-ICS being significantly faster than that of WPS-ICS, is not visible in Figure 1.

Response: Thanks for your comments. The relative calculated results can be found in Table1. We have revised the citation form here.

Both Figure 1 and 3 does not have figure legend for WPS-ICI and WPS-ICS.

Response: Thanks for your comments. We have added the figure legend for WPS-ICI and WPS-ICS.

Furthermore, the error bars seem to be off, e.g. the max and min SD are not of the same length. Why is that? It is also not visible in some figures what stepwise increase in irradiance mean (shade to white) - please direct more the readers on this.

Response: Thanks for your comments. All error bars have been shown in the figures.  As the points are recorded individually during induction, all SD are different. All shaded parts are recorded for approximately 2 minute. 

I also suggest that the authors switch Figure 3 and Figure 4, as they first described Figure 4 in the Results section, this figure should be renamed as Figure 3 instead, and vice versa. This is common practice.

Response: Thanks for your comments. We have changed the order of the results part.

During the discussion, I would suggest that the authors would first translate what the treatments mean in physiological terms (i.e. instead of Illumination and Shading, use low light and high light, respectively). This will create a more targeted discussion and would be more easy to understand by the readers.

Response: Thanks for your comments. We are trying to make the results more easy to understand. The original versions did use low light and high light to describe the treatments, but reviewers thought its confusing since the measurements of photosynthetic induction is also from low light to high light.

Moreover, once the authors have described the results more correctly, they can adjust their discussion later on. Currently, there are parts of the discussion that is not very visible in the results (e.g. line 346 to 347, line 351 to 352 and line 367 to 370).

Response: Thanks for your comments. We have revised these sentences.

Other minor comments:

I suggest refining more the abstract, the hypothesis does not seem to be compatible with the method. Please refine the overall hypothesis by summarising the last 3 questions in the end of introduction, and bring that to the abstract.

Response: Thanks for your comments. We have rewrite the hypothesis in the introduction and abstract parts.

Line 30, nice sentnce.

Line 36, use a better conjunction. Instead of "As a matter of fact", perhaps use, "However, in reality,"... This will connect sentence 34-35 to sentence 36-37

Response: Revised.

Line 39-40, needs reference

Response: Added.

Line 46-47: IRGA chamber Shading is just low light condition in the method, I suggest to use low light instead of shading, as they would mean different things, especially with the spectral quality.

Response: Thanks for your comments. We have unified all the treatments throughout literature.

Line 59-60: Good content, but citation is confusing. Why cite Acevedo-Siaca et al 2021, if the RUBISCO carboxylation limitation is showed in their 2020 paper?

Response: Revised.

Line 66-68, I think there are already research whether stomatal and biochemical limitation affected by low and high light during photosynthetic induction, so we know a lot about it (in several crop species).

Response: Not really. It’s true that several studies have indicated the limitations of stomatal and biochemical processes to photosynthetic induction. However, the effect of IRGA Chamber Illumination and Whole Plant Illumination on stomatal and biochemical limitations is still not well understood.

Line 94: put s in month

Response: Revised.

Line 117 to 131: I suggest to make a schematic diagram to illustrate the changes in light intensity for each treatment. This will give the reader a better grasp of the method.

Response: Thanks for your comments. We have showed the response curve of gas exchange to a stepwise increase in light intensity across all treatments in Figure 1. Another schematic diagram may overlap with Figure 1.

Line 179-181, Figure 1 shows the contrary. Please be more careful with how you describe your results. This makes your discussion difficult to trust if your results are not correctly described.

Response: Revised.

Line 184 - 185, this is not visible in Figure 1, (same with line 186-187)

Response: Revised.

Line 191-192: It seems that lower Ci at the end of induction in ICI than ICS is not significant. Can you test?

Response: Yes, it’s only significant at the initial phase. We don’t mention significant differences here.

Line 213-216, please point the reader to which Figure.

Response: Revised.

Line 216, I dont see this in Table 1. Can you please reassess?

Response: P50 and P90 of WPS-ICS showed the highest values than other treatments.

Line 232-233: This was not indicated in the results or supplementary figures?

Response: Table S1. Revised.

Comments on the Quality of English Language

I suggest editing the paper for english correction. Lots of parts of the manuscript had wrong grammar (for example, line 14, line 52, line 94, so on..)

Response: Thanks for your comments. We have carefully revised the grammar of the literature.

Round 2

Reviewer 3 Report

Comments and Suggestions for Authors

Dear Authors,

Thank you for the responses. I regret to say that I still find the paper lacking the quality and substance to merit publication. I hope the authors will use my comments to guide them in better describing their results to have a manuscript that is worth publishing in your MDPI Plant Journal.

reviewer

Comments on the Quality of English Language

There are still minor typo error. Please proofread the entire manuscript for this.

Author Response

please check the revision